# Investigation of clinical characteristics and genome associations in the 'UK Lipoedema' cohort

Dionysios Grigoriadis[1]⊚, Ege Sackey[1]⊚, Katie Riches[2]‡, Malou van Zanten[1]‡, Glen Brice[3], Ruth England[2], Mike Mills🄳[1], Sara E. Dobbins🄳[1], Li Ling Lee[1], Lipoedema Consortium[¶], Genomics England Research Consortium[¶], Steve Jeffery[1], Liang Dong[4], David B. Savage[4], Peter S. Mortimer[1,5], Vaughan Keeley[2,6], Alan Pittman[1], Kristiana Gordon[1,5]*, Pia Ostergaard🄳[1]*

1 Molecular and Clinical Sciences Institute, St George's University of London, London, United Kingdom, 2 University Hospitals of Derby and Burton NHS Foundation Trust, Derby, United Kingdom, 3 South West Thames Regional Genetics Unit, St George's University of London, London, United Kingdom, 4 Metabolic Research Laboratories, Wellcome Trust-Medical Research Council Institute of Metabolic Science, University of Cambridge, Cambridge, United Kingdom, 5 Dermatology & Lymphovascular Medicine, St George's Universities NHS Foundation trust, London, United Kingdom, 6 University of Nottingham Medical School, Nottingham, United Kingdom

⊚ These authors contributed equally to this work.
‡ KR and MZ also contributed equally to this work.
¶ See acknowledgement for more information about the consortium.
* kristiana.gordon@stgeorges.nhs.uk (KG); posterga@sgul.ac.uk (PO)

**Data Availability Statement:** The discovery data are available on Figshare (DOI: 10.24376/rd.sgul. 16803109).

## Abstract

Lipoedema is a chronic adipose tissue disorder mainly affecting women, causing excess subcutaneous fat deposition on the lower limbs with pain and tenderness. There is often a family history of lipoedema, suggesting a genetic origin, but the contribution of genetics is currently unclear. A tightly phenotyped cohort of 200 lipoedema patients was recruited from two UK specialist clinics. Objective clinical characteristics and measures of quality of life data were obtained. In an attempt to understand the genetic architecture of the disease better, genome-wide single nucleotide polymorphism (SNP) genotype data were obtained, and a genome wide association study (GWAS) was performed on 130 of the recruits. The analysis revealed genetic loci suggestively associated with the lipoedema phenotype, with further support provided by an independent cohort taken from the 100,000 Genomes Project. The top SNP rs1409440 ($OR_{meta} \approx 2.01$, $P_{meta} \approx 4 \times 10^{-6}$) is located upstream of LHFPL6, which is thought to be involved with lipoma formation. Exactly how this relates to lipoedema is not yet understood. This first GWAS of a UK lipoedema cohort has identified genetic regions of suggestive association with the disease. Further replication of these findings in different populations is warranted.

## Introduction

Lipoedema is a chronic condition characterized by abnormal subcutaneous accumulation of adipose tissue in the limbs. This condition predominantly affects women, and the clinical

**Funding:** This project was supported by Lipedema Foundation (https://www.lipedema.org/) LF#006 (KG and PO), the Wellcome Trust (https://wellcome.org/) WT107064 (DBS), and the MRC Metabolic Disease Unit, and The National Institute for Health Research (NIHR) Cambridge Biomedical Research Centre and NIHR Rare Disease Translational Research Collaboration (https://www.mrl.ims.cam.ac.uk/mrc-metabolic-diseases-unit/) MRC_MC_UU_12012.1 (DBS). The funders had no role in study design, data collection and analysis, decision to publish, or preparation of the manuscript.

**Competing interests:** The authors have declared that no competing interests exist.

phenotype is of a disproportionate figure with symmetrically enlarged lower body, typically affecting the hips and buttocks, extending to the legs, with sparing of the feet leading to a bracelet or cuffing effect at the ankles. Some patients have a more proximal distribution of abnormal fat, with the thighs affected to a greater extent than the lower legs. The affected tissues feel soft and "doughy" to the touch and the skin remains soft unlike in lymphoedema. In some patients the abnormal fat is reported to feel grainy, nodular or like "beans in a bag" [1, 2]. The torso appears unaffected, and, in the absence of obesity, individuals present with a relatively small waist and chest. The upper limbs may also be involved, but with forearm sparing in many cases. The onset of lipoedema often occurs at times of female hormonal change such as puberty, during pregnancy or menopause [3, 4]. The condition is associated with easy bruising, tenderness when touched, and chronic pain in the affected limbs [5]. The pain is frequently misdiagnosed as fibromyalgia. Chronic fatigue, psychosocial and poor body image issues are recognized comorbidities with lipoedema.

The term lipoedema itself causes confusion amongst medical professionals. Whilst it is derived from Latin and Greek words for "fat" (lipid or lipos) and "to swell" (oedema or oidein), physicians tend to use the term "oedema" in clinical practice to refer to the presence of fluid swelling. Lipoedema remains largely underdiagnosed or even misdiagnosed by the medical profession [6, 7]. One explanation for diagnostic difficulties is that lipoedema is a little-known disease, which can also be confused diagnostically with other conditions that present with limb enlargement such as lymphoedema or gynoid obesity [3, 8]. For these reasons, there is a paucity of prevalence estimates, some studies report an estimated prevalence of 10–11% while others suggest 1 in 72,000 which is likely an underestimate due to misdiagnosis in the community [3, 9, 10].

In chronic lymphoedema there can be a significant fat composition which contributes to leg swelling [11] and secondary lymphoedema may complicate lipoedema, so called lipolymphoedema. Another distinguishing feature of lymphoedema is a high rate of cellulitis due to immune dysfunction from impaired lymphatic drainage [12]. This again contrasts with an absence of cellulitis reports in lipoedema patients (unless they have developed secondary lymphoedema). Furthermore, a histological and molecular characterisation of skin and fat biopsies identified no related lymphatic anomaly in lipoedema patients strengthening the argument for distinct aetiologies [13].

Lipoedema is not always simple to differentiate from obesity. Gynoid fat distribution can look identical to lipoedema but is less painful and in theory responsive to calorie restriction. Obesity may be assessed by calculating body mass index (BMI), defined as the weight in kilograms divided by the square of the height in meters (kg/m$^2$). The WHO categorises a BMI over 25 kg/m$^2$ as overweight, and a BMI over 30 kg/m$^2$ as obese [14]. Patients with lipoedema usually have elevated BMIs because of big heavy legs but whilst obesity will respond to restricted dietary intake, the abnormal fat of lipoedema is far less responsive, leading to a wasted upper body but a lower body that stubbornly remains disproportionately enlarged from the waist to the ankles. The abnormal response to weight-reducing diets would argue against a form of obesity. However, later in life, lipoedema is often complicated by obesity, in which case, historical symptoms of disproportionately big legs but small upper trunk are key to the diagnosis. Interestingly, patients with lipoedema display a less severe cardiovascular profile and have a normal lipid profile than those of equivalent BMI without lipoedema [15–17]. The gynoid profile of lipoedema may even protect against diabetes [15, 18].

The diagnosis of lipoedema can be difficult to make if lymphoedema and/or obesity co-exist. One of the major problems with the diagnosis of lipoedema is the lack of a confirmatory test. The exploration of ultrasound in lipoedema proves promising [9, 19]; however, it is not yet an established gold standard.

Exactly what causes lipoedema is not known. Family history has been reported in lipoedema patients suggesting a familial origin of the disease [5, 20] but many cases also appear to be sporadic. Genes or loci associated with the condition are still in need of identification. With only one report on a single gene (monogenic) cause in a single family [21], we believe that the genetic architecture of the disease is more complex with a mix of genetic and environmental risk factors contributing. To investigate this hypothesis, we have conducted a Genome Wide Association Study (GWAS) to investigate genetic associations with the lipoedema trait. Obtaining meaningful genetic results relies on studying as homogenous a group of phenotypes as possible. Therefore, the cohort of patients were selected on strict clinical criteria. As reduced quality of life has been reported in women with lipoedema [4, 22], the recruits were also subjected to self-administered health related quality of life (HRQoL) assessment as the items that are objectively measured in the HRQoL assessment can assist in the diagnostic criteria. Identifying the possible genetic causes could help to better define lipoedema, facilitate the development of a diagnostic test, and could lead to possible treatments.

## Materials and methods

### Case ascertainment

Patient recruitment occurred through referrals to the two UK specialist clinics at St George's University Hospital NHS Trust and the University Hospitals of Derby and Burton NHS Trust. Further recruitment was encouraged through advertisement to the members of 'LipoedemaUK' local patient support group meetings and conferences. The patients were seen by clinicians or a research nurse with a specialist interest in lymphoedema and lipoedema (authors GB, KG, KR, PSM, RE and VK). Ethical approval was obtained from the local Health Research Authority (Fulham NRES Committee, London; REC reference number: 16/LO/0005). Individuals were invited to participate if they matched the major inclusion criteria and written informed consent was obtained from all participants. Methodological details regarding inclusion criteria and the data obtained through interview and clinical assessment are reported in Table 1 and the Supplementary Methods (S1 File).

### Health related quality of life assessment

Patients were invited to complete the 36 items of the General Health Questionnaire Short Form (SF-36 Health Survey) by themselves at the time of their appointment. Participation in this study was voluntary and no incentives were offered. The SF-36 measures eight domains related to 'Physical Functioning', 'Physical Role Limitations', 'Emotional Role Limitations', 'Vitality' (or energy), 'Emotional and Mental Wellbeing', 'Social Functioning', 'Bodily Pain',

**Table 1. Summary of inclusion criteria.**

| Inclusion criteria |
|---|
| Female |
| Age of onset (<35y) |
| BMI $\leq$40 kg/m$^2$ |
| Waist-hip ratio (WHR) $\leq$0.85 |
| No or minimal central (android) obesity |
| Bilateral and symmetrical fat hypertrophy on lower limbs |
| Spared feet |
| Persistent enlargement (with no significant effect from overnight elevation) |
| White British ethnicity (only for the GWAS) |

and 'General Health'. If more than 25% of the questionnaire was incomplete, it was excluded from the analysis. The Likert like scores were transformed to range from zero to 100 and the methods for computing the scores followed reported guidelines [23]. Scores from the eight SF-36 domains were correlated with clinical records such as participant BMI, age of onset of lipoedema and waist-hip ratio (WHR). Correlations between the SF-36 domain scores, and between demographic variables were also computed.

## Genotyping of the discovery cohort

Recruits who identified as being of white British ancestry were invited to participate in the genotyping arm of the study. 148 consented and peripheral blood was obtained, DNA extracted, and genotyped in two batches by Cambridge Genomic Services using Illumina Infinium_CoreExome-24_v1-2 single nucleotide polymorphism (SNP) chip and by UCL Genomics facilities using the Infinium_Core-24_v1-2-a1 SNP chip. To avoid batch effect generated by genotyping the lipoedema samples in two slightly different SNP arrays, 22 samples were genotyped in both batches and after stringent QC including call rate and HWE analysis SNPs showing inconsistency (n = 4) between the two arrays were removed. 5,849 female samples of white British ethnicity enrolled in the Understanding Society UK study [24] and genotyped using HumanCore Exome-12_v1.0 were used as controls (European Genome-phenome Archive ID: EGAD00010000890).

## Replication cohort

For the replication study, whole genome sequencing data from the Genomics England (GEL) 100,000 Genomes Project Rare Diseases program (main-programme_v11) was used [25]. In the Cardiovascular Genomics England Clinical Interpretation Partnership (GeCIP), 93 participants were identified with the label "Lipoedema" in the lymphatic disorder subdomain. To ensure there was no overlap between the discovery cohort and the replication cohort, GEL participants already included in the discovery cohort or participants related to individuals in the discovery cohort were excluded. GEL participants not marked as "Europeans" by the 100,000 Genomes Project inferred ancestry were also excluded, and so were individuals who had HPO terms indicating comorbidities unrelated to lipoedema, leaving us with 27 cases for the replication cohort. Unaffected females without a diagnosed condition, marked as "Europeans" and unrelated to each other and/or to the lipoedema cases were selected as the control group of the replication cohort (N = 11,409).

Nineteen samples with SNP chip genotypes were also available as whole genome sequencing data from the GEL Project. Comparison of selected SNPs (n = 6, top SNPs from replication) identified a concordancy of 100% between platforms.

## Association analyses and meta-analysis

Discovery cohort and control genotyping data underwent thorough quality control before association analysis using PLINK (v1.90b6.21 & v2.00a3LM) [26]. Samples with either low calling rate (< 97%) or ±3 SD deviation from the heterozygosity rate mean of the samples ($N_{Cases}$ = 4, $N_{Controls}$ = 85) were excluded from the analysis. All samples were confirmed as female using the PLINK sexcheck function (F inbreeding coefficient, <0.2 for females). Relatedness between all sample pairs in the cohort was inferred by calculating identity by descent. In sample-pairs with PI_HAT>0.05, the sample with the highest BMI and WHR (for cases) and/or lower genotyping calling rate was excluded ($N_{Cases}$ = 9, $N_{Controls}$ = 304). The cohort was merged with the CEU, CHB and YRI reference populations from HapMap study [27] and genetically divergent ethnic outliers were excluded ($N_{Cases}$ = 5, $N_{Controls}$ = 59) after performing

principal component analysis using GCTA package (v1.93.2beta) [28] leaving 130 cases and 5,401 controls in the discovery cohort. SNPs with minor allele frequency (MAF) <0.05, missing call rate >0.05, and Hardy-Weinberg equilibrium $\leq 1 \times 10^{-6}$ were excluded from the analysis.

SNP-based heritability association analysis was then calculated in the discovery cohort by using the restricted maximum likelihood (—reml) option in the GCTA package. Since the prevalence of lipoedema is still elusive the calculation was performed by using both a prevalence of 5% and 10%.

Association analysis was performed in the discovery cohort using PLINK 1.9 logistic regression. The distribution of the association *P*-Values was assessed using a Quantile-Quantile plot (Q-Q) plot. The 26 SNPs with the lowest *P*-values in 23 distinct loci were tested for association with lipoedema in the replication cohort using PLINK 1.9 logistic regression. Summary statistics from both studies were used to perform a meta-analysis for these 26 SNPs using METAL software [29]. The "SCHEME STDERR" approach was followed so the meta-analysis was performed on Odds Ratios (OR) and their standard errors. These SNPs were annotated using SNPnexus [30], while their impact on gene expression in different tissues was explored using expression quantitative trait loci (eQTL) information from the Genotype Tissue Expression (GTEx) Portal v7 [31] using LDexpress Tool [32] ($r^2$ >0.6, European Population). Only results with P<6x10$^{-8}$ are reported as significant (based on Bonferroni correction of 948 SNPs tested across 54 tissue types for genes within 500kb of lead SNPs). Due to the highly correlated nature of the tests all results with P<0.001 are included in Supplementary Tables for reference.

Colocalization analysis was performed using LocusFocus (version 1.4.9) [33]. LocusFocus implements the Simple Sum (SS) colocalization method based on a frequentist framework developed by Gong et al. [34]. LocusFocus presents the degree of colocalization of genes across the tissues with a -log10 Simple Sum P-value (SSP). A conservative approach for multiple testing was adopted by implementing a Bonferroni based correction for identifying significantly colocalized signals based on number of gene-tissue pairs tested (SSP>3.2). Genomic regions of interest were defined as ±250 kb from the most associated SNP per locus, or ±750kb for regions with a low recombination rate.

Further SNP fine mapping was performed by using the ENCODE Candidate Cis-Regulatory Elements combined from all cell types [35] and Clustered interactions of GeneHancer regulatory elements and genes [36] databases using the UCSC Table Browser [37]. Scripts used for the analysis can be found on GitHub (https://github.com/digrigor/SGUL_UK_Lipoedema_GWAS).

## Results

### Patient selection criteria

Patient selection used clearly defined clinical criteria (Table 1, S1 File). These included painful excess adipose deposition from the hips to the ankles (Fig 1A–1G), BMI ≤ 40 but no excess upper body fat, waist-hip ratio (WHR) ≤ 0.85, soft and "doughy" tissues, and sparing of the feet. Women with proximal upper limb lipoedema (Fig 1G) were also included in the study.

Additional patients were included who might not have had a clear-cut diagnosis. One patient was initially diagnosed with lower limb lymphoedema as a result of morbid obesity. Bariatric surgery was undertaken, and significant weight loss was achieved (~50kg reduction). Her four-limb lipoedema had been masked by the obesity and only became apparent after the weight loss revealed disproportionate fat deposition of the limbs (Fig 1H–1J). Other patients presented with BMI > 40 at time of recruitment to the study (Fig 1K–1M), but as they were longstanding patients of the clinic with documentation of BMI < 35 at the time of

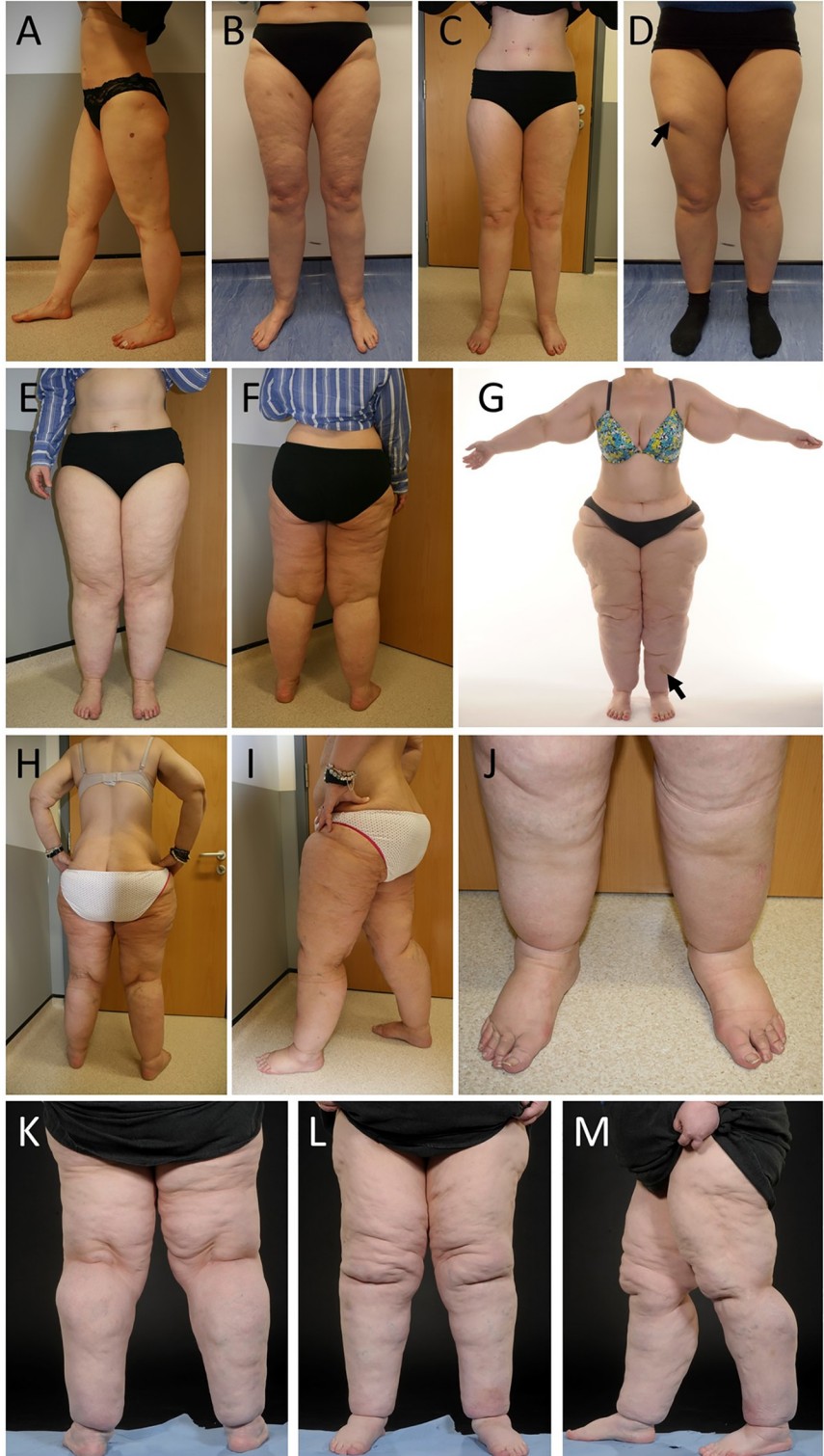

**Fig 1. Examples of recruited patients.** (A-C) Three female patients with relatively mild lower limb lipoedema manifesting with excess adipose deposition from the hips to the ankles. The patients do not have obesity and their BMIs range from 23.7 (within the normal/healthy weight range) to 26.6 (overweight). Waist-hip ratios (WHR) for all three women are less than 0.75. (D) A female patient with lower limb lipoedema possessing the same clinical signs as patients in A-C. However, the additional finding of a well-defined lipoma is clearly visible on the right anterior thigh

(arrow). (E-G) Two women with moderately severe lower limb lipoedema. Proximal upper limb lipoedema is noticeable in (G). Ankle braceleting is clearly present in both women, and there is no evidence of secondary lymphoedema or venous disease. Both women have an elevated BMI in the "obesity" range, but their WHRs are less than 0.75. There is a bruise on the left shin in (G) (arrow) that reportedly developed after minimal trauma to the area. (H-J) A female patient with four-limb lipoedema, mild lymphoedema of the lower legs and grade CEAP2 venous disease with telangiectasia and asymptomatic varicose veins. This patient was initially diagnosed with lower limb lymphoedema as a result of morbid obesity. Bariatric surgery was undertaken, and significant weight loss was achieved (~50kg). Her four-limb lipoedema had been masked by the obesity and only became apparent after significant weight loss revealed disproportionate fat deposition of the limbs. (K-M) This patient with severe lower limb lipoedema did not meet initial inclusion criteria as her BMI was 44 at the time of recruitment, despite a WHR of 0.78. However, she is a longstanding patient of the clinic with documentation of BMI <35 at time of first presentation.

presentation, they were included. Although these recruits had increased waistline and android fat distribution, all still had a WHR ≤ 0.85 and the significantly elevated BMI reflects the progression of lipoedema over several years with increasing volumes of disproportionate (gynoid) adipose deposition of the lower limbs.

## Patient summary characteristics

A total number of 200 women were recruited between September 2016 and March 2018 through face-to-face interview and clinical examination. The face-to-face interview included questions that are often self-reported by women with lipoedema such as the presence of pain or tenderness to the touch, noticeable easy bruising and disproportionate weight loss upon dieting. A summary of patient characteristics is documented in Table 2 and the full data are available in S1 Table.

At recruitment, the majority stated they were white British (92.5%), and the mean age was 47 years (SD±13.5; range 18y-81y) (Table 2). On average, the individuals reported to have been affected by lipoedema for 29.2 years (SD±12.9) with an age of onset at 16.8 years old (SD ±9.0).

Clinical examination showed the mean weight among the lipoedema cases was 90.4kg (SD ±20.0), mean height 1.65m (SD± .07) and the mean BMI was 33.4 (SD±7.2) (Table 2). The high BMI was not due to high levels of android fat as the average waist circumference was 91.3cm (SD±13.4), and hip circumference was 120.4cm (SD±14.3), thus the average calculated waist-hip ratio (WHR) was 0.76 (SD±0.07). This is less than the WHO recommended WHR of 0.85 for women, indicating that central obesity was not the cause of elevated BMI values [38]. Distribution of BMI, WHR and waist circumference among the cases are shown in S1 Fig.

Patients were examined for hypermobility or joint laxity of the elbows, knees, small joints of the hands and the back during the clinical assessment because there are anecdotal reports of increased hypermobility with lipoedema. 17.8% (33 recruits out of 185) were hypermobile (Table 2). Individuals were also examined for the presence of pitting oedema as part of the clinical assessment. In 53 (27%) recruits, mild pitting oedema was observed. The majority of oedema was observed in the older age groups (49/53 individuals with oedema were >35y). In most cases the oedema was either intermittent or confined to the ankles (61.7%). The underlying reason for the oedema was not investigated.

The face-to-face interview revealed that 58.2% (110/189) self-reported to have a family history of lipoedema (Table 2). Easy bruising, seen as one of the parameters to assess lipoedema, was self-reported in 90.3% individuals. 71% reported their limbs to be tender to the touch. On examination, 47.4% had clinically evident venous abnormalities, mostly mild superficial telangiectasia or uncomplicated varicose veins consistent with CEAP C1 and C2 disease (Table 2, Fig 1H–1M).

**Table 2. Summary characteristics of the 'UK Lipoedema' cohort.** Observations from the clinical examination and information obtained through face-to-face interview at time of recruitment is included. A total of 200 individuals were recruited to the study. 'Missing data' indicates the number of individuals where values were not obtained.

| | Mean ± SD | Range | Missing data |
|---|---|---|---|
| Age at evaluation (yrs) | 47.0 ± 13.5 | (18–81) | 0 |
| **Age at evaluation classes (yrs)** | **N** | **%** | |
| 18–25 | 12 | 6.0 | |
| 26–35 | 25 | 12.5 | |
| 36–50 | 80 | 40.0 | |
| 51–65 | 65 | 32.5 | |
| >65 | 18 | 9.0 | |
| | **Mean ± SD** | **Range** | **Missing data** |
| Age at onset of lipoedema (yrs)* | 16.8 ± 9.0 | (6–60) | 22 |
| Start of puberty (yrs)* | 12.5 ± 1.6 | (9–17) | 18 |
| Disease duration (yrs) | 29.2 ± 12.9 | (1–62) | 22 |
| Height (m) | 1.65 ± 0.07 | (1.46–1.85) | 5 |
| Weight (kg) | 90.4 ± 20.0 | (47–160) | 9 |
| BMI | 33.4 ± 7.2 | (19.0–58.8) | 8 |
| Waist circumference (cm) | 91.3 ± 13.4 | (42–123) | 38 |
| Hip circumference (cm) | 120.4 ± 14.3 | (90–169) | 37 |
| Waist-hip ratio (WHR) | 0.76 ± 0.07 | (0.40–0.93) | 38 |
| **BMI class** | **N** | **%** | |
| <25 | 21 | 10.8 | |
| 25–29.9 | 48 | 25.0 | |
| 30–34.9 | 48 | 25.0 | |
| 35–39.9 | 42 | 21.9 | |
| 40–49.9 | 30 | 15.6 | |
| ≥50 | 3 | 1.6 | |
| | **%** | **(N/total)** | **Missing data** |
| White British* | 92.5 | (185/200) | 0 |
| Family history* | 58.2 | (110/189) | 11 |
| **Oedema** | | | |
| **Age class (yrs)** | **N with oedema** | **Total** | **%** |
| ≤35 | 4 | 35 | 11.4 |
| 35–60 | 35 | 126 | 27.8 |
| >60 | 14 | 35 | 40.0 |
| **All oedema** | **53** | **196** | **27.0** |
| | **%** | **(N/total)** | **Missing data** |
| *Oedema of ankle (sometimes incl feet)* | *51.1* | *(24/47)* | *6* |
| *Oedema of leg/lower limb* | *38.3* | *(18/47)* | *6* |
| *Intermittent oedema** | *10.6* | *(5/47)* | *6* |
| **Venous problems‡** | **%** | **(N/total)** | **Missing data** |
| *CEAP ≥C3* | 2.6 | (5/190) | 10 |
| *CEAP C2* | 25.3 | (48/190) | 10 |
| *CEAP C1* | 19.5 | (37/190) | 10 |
| *CEAP C0* | 52.6 | (100/190) | 10 |
| Venous problems and lymphoedema | 13.2 | (25/190) | 10 |
| **Other features** | **%** | **(N/total)** | **Missing data** |
| Tender to touch and pain* | 71.0 | (110/155) | 45 |
| Bruise easily* | 90.3 | (139/154) | 46 |

(*Continued*)

**Table 2.** (Continued)

| | | | |
|---|---|---|---|
| Hypermobility | 17.8 | (33/185) | 15 |
| Pes planus | 22.2 | (40/180) | 20 |
| Liposuction* | 6.2 | (11/177) | 23 |
| Responsiveness to dieting* | | | |
| *Disproportional response* | 86.7 | (144/166) | 34 |
| *No loss* | 7.8 | (13/166) | 34 |
| *Equal loss all over* | 5.4 | (9/166) | 34 |

*, self-reported, information obtained through interview; N, number of cases; SD, standard deviation; Total, total number of cases; yrs, years. Disease duration calculated as 'Age at evaluation' minus 'Age at onset of lipoedema'.

*, self-reported, information obtained through interview; N, number of cases; SD, standard deviation; Total, total number of cases; yrs, years.

‡We were not specifically assessing for varicose veins; no venous duplex was carried out, so hidden (deeper) varicose veins have not been recorded. CEAP, Clinical Etiological Anatomical Pathophysiological classification; CEAP C0, no visible or palpable varicose veins; CEAP C1, mild superficial venous problems such as spider, reticular or thread veins (telangiectatic); CEAP C2, uncomplicated varicose veins such as enlarged, prominent veins; CEAP >C3, varicose veins with symptoms.

When asked about the effect of dieting, 86.7% of recruits reported a disproportional weight loss where they found it easier/quicker to lose weight from the torso compared to the limbs (Table 2). Only 7.8% reported no loss of fat at all from the limbs with dieting/weight loss. 6.2% of recruits had undergone liposuction and one individual had undergone bariatric surgery. The bariatric surgery had led to 50kg weight loss, but unfortunately this accentuated her disproportionate body shape as more weight was lost from the torso compared to the limbs unmasking the lipoedema phenotype (Fig 1H and 1I).

## Health related quality of life assessment

Physical, social and mental aspects of health were evaluated using the validated and widely used self-reported Short Form-36 Quality of Life Questionnaire (SF-36). 135 women of the 200 recruited completed enough domains of the questionnaire to be included for analysis (S2 Table). The scores across the eight domains ranged from 40.2–64.7 (out of 100; with 100 indicating better health status) (Table 3). Multiple significant ($P < 0.05$) correlations were found between SF-36 scores and clinical variables (S3 Table). The strength of most of the

**Table 3. Outcome of evaluation of health status in 135 lipoedema cases who completed the SF-36 quality of life questionnaire.**

| Dimension | Mean | Std. Deviation |
|---|---|---|
| Physical functioning | 61.1 | 28.0 |
| Role limitations physical | 58.9 | 42.9 |
| Role limitations emotional | 57.9 | 42.9 |
| Vitality | 40.2 | 23.9 |
| Emotional/mental Wellbeing | 60.1 | 19.9 |
| Social functioning | 64.7 | 27.1 |
| Bodily pain | 57.1 | 27.1 |
| General health | 49.5 | 21.2 |

The mean score and the standard deviation of all dimensions is given. The individual 36 questions are scored with a Likert-type scale and the eight domains of health are computed means to a 0–100 scale. Higher scores indicate better health status.

relationships was weak-moderate as the absolute value of the correlation coefficient, $r$, was < 0.7. A few domains did show a strong relationship with the bodily pain domain, so that those experiencing lots of pain in the bodily pain domain also would report worse general health ($r = 0.70$) and physical functioning ($r = 0.78$) (S3 Table). The social functioning domain was also found to correlate strongly to the emotional and mental wellbeing domain ($r = 0.72$).

## Genome-wide association analysis

Of the 200 recruited lipoedema cases, 130 white British were included in a GWAS discovery cohort (indicated in S1 Table) with 5,531 ethnically matched female controls from the Understanding Society the UK Household Longitudinal Study cohort. The replication cohort consisted of 27 ethnically matched lipoedema cases (S4 Table) and 11,409 female controls enrolled in the 100,000 Genomes Project Rare Diseases Program v11. After quality control implementation, 233,441 SNPs were tested for association with the lipoedema trait in the discovery cohort using logistic regression analysis. The 26 SNPs showing the greatest association with lipoedema were selected for replication in the independent cohort, where genotyping was done by Whole Genome Sequencing, using logistic regression analysis. A meta-analysis was then performed to pool the per-SNP effect sizes from the discovery and replication studies.

To ensure there was no systematic bias in the discovery study arising from population stratification, a principal component analysis was performed with the HapMap population reference panel samples, revealing that after the quality-control steps there are no ethnic outliers left in the study, as both lipoedema cases and controls cluster together with the Central European HapMap population (Fig 2A). This is further highlighted by the absence of genomic inflation ($\lambda_{gc} = 1.004$) on the QQ plot of the observed $P$ values (Fig 2B). To understand the proportion of genetic variance influencing the lipoedema phenotype in our cohort, SNP-based heritability ($h^2_{SNP}$) in the discovery cohort was estimated and found to be 0.50 (SE = 0.52, $P = 0.17$) and 0.62 (SE = 0.65, $P = 0.17$) when the prevalence of lipoedema in the population was set to 5% and 10%, respectively. However, there is a lack of statistical significance in this estimation due to the limited sample size.

The association analysis in the discovery cohort revealed multiple suggestive genomic loci associated with lipoedema ($P < 2 \times 10^{-4}$). Although there were no SNPs passing the genome-wide significance threshold ($P < 5 \times 10^{-8}$), 26 SNPs with $P < 2 \times 10^{-4}$ were identified (Fig 2C, S5 Table, MAF ≥0.05). Six of these SNPs (in four distinct loci) were supported in the replication cohort with $P_{meta} < 1 \times 10^{-4}$ and same direction of effect for both analyses (Table 4).

According to the meta-analysis, the top three lipoedema-associated SNPs (rs1409440, rs7994616, and rs11616618; $OR_{meta} \approx 2.01$, $P_{meta} \approx 4 \times 10^{-6}$, Fig 2D) are in a block of linkage disequilibrium (LD) on chromosome 13. The block, which is ~40kb with $r^2 > 0.8$, is near the *FREM2*, *STOML3*, *PROSER1*, *NHLRC3*, and *LHFPL6* genes (Fig 2E). When mapping these non-coding SNPs to regulatory elements in the genome all three are located in an *LHFLP6* interaction region according to the GeneHancer database, while based on the ENCODE project classifications, rs1409440 is specifically located in a distal enhancer-like signature locus upstream of *LHFPL6* (S6 Table). Localization of this LD block in regulatory elements of *LHFPL6* suggests it is a regulator of the gene's expression. Interestingly, we observe some evidence of association between several LD buddies ($r^2 > 0.6$) of the three SNPs with *LHFPL6* expression through expression quantitative trait loci (eQTLs) analysis ($P < 5 \times 10^{-6}$, S7 Table).

To explore whether the presence of these three variants upstream of *LHFPL6* affects the clinical characteristics of the carriers, the phenotypic characteristics of the group of patients carrying all three SNPs (N = 45: $N_{Discovery} = 38$, $N_{Replication} = 7$) were compared against those of non-carriers in both discovery and replication cohorts. The results showed that lipoedema

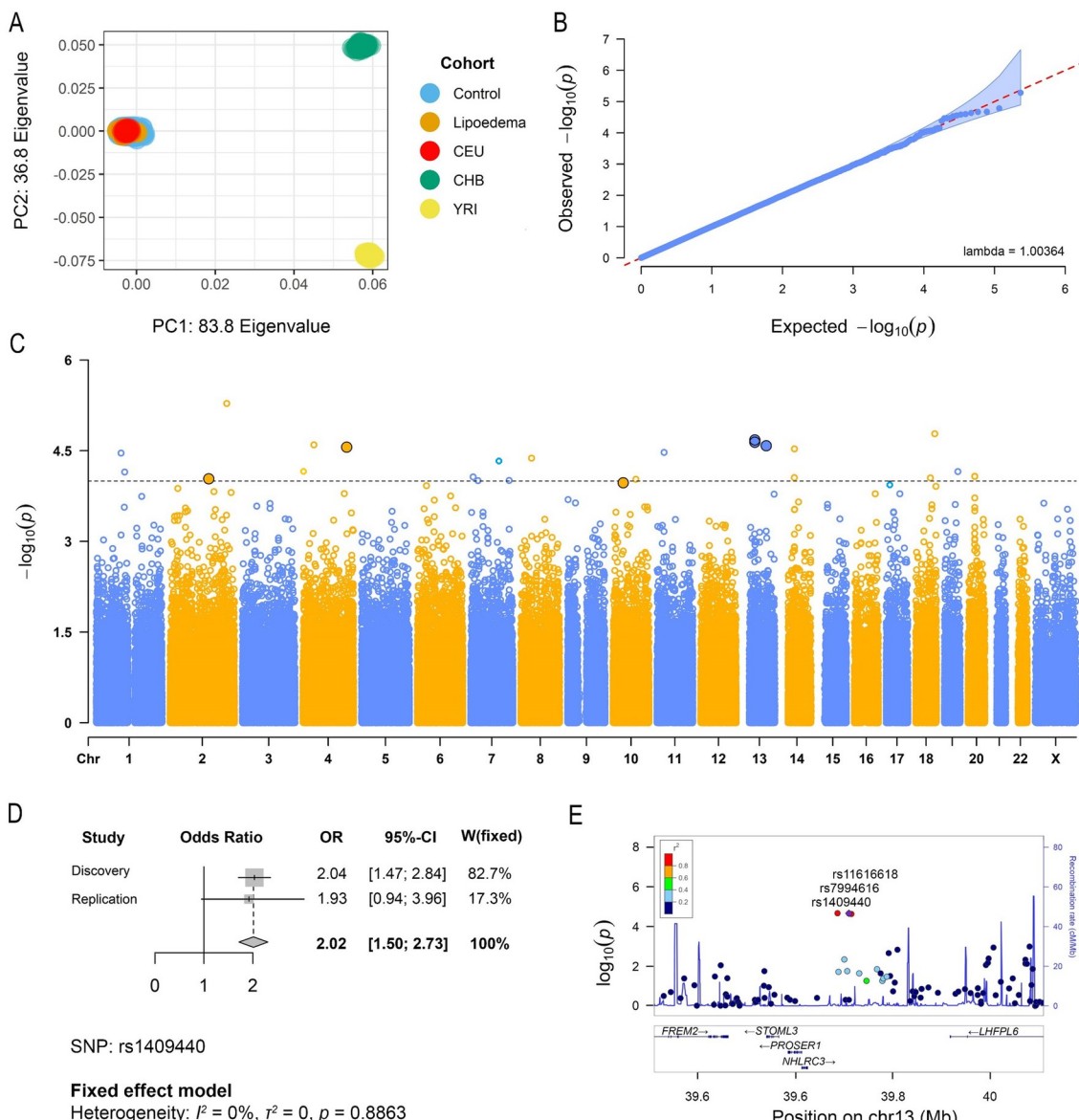

**Fig 2. Results of the 'UK Lipoedema' discovery cohort Genome-Wide Association Study (GWAS) and meta-analysis.** (A) Plot of the first two principal components from the PCA performed on the GWAS (lipoedema cases and controls) samples and the HapMap individuals. (B) Quantile–Quantile plot of GWAS samples showing no genomic inflation. (C) Manhattan plot of the genome-wide $P$-values (in–$\log_{10}$ scale) of association with lipoedema in the discovery cohort. The association was tested using logistic regression analysis. The highlighted SNPs (dots with black outline) were tested in the replication cohort and have $P_{meta} < 1 \times 10^{-4}$ and same direction of effect for both studies. (D) Forest plot of the chromosome 13 rs1409440 SNP meta-analysis pooled odds ratios and 95% confidence intervals. (E) Regional plot of the top three SNPs (rs1409440, rs7994616, rs11616618) from the meta-analysis in one distinct genomic locus on chromosome 13 near the *LHFPL6* gene.

patients carrying the variants upstream of *LHFPL6* were significantly more likely to report a direct maternal relative (mother, daughter, sister) with lipoedema symptoms (chi-squared test: $\chi^2$ (1, N = 157) = 10.03, $P = 0.002$), highlighting the putative contribution of this locus, upstream of *LHFPL6*, to the genetic aspect of the disease.

Next, we explored the eQTL signals of the other SNPs (Table 4), to investigate links with the lipoedema phenotype. The SNP rs11511253 ($P_{meta} = 4.07 \times 10^{-5}$, $OR_{meta} = 1.67$) is

**Table 4. List of the meta-analysis replicated variants.**

| | | | | | | Discovery | | | Replication | | | Meta Analysis | |
|---|---|---|---|---|---|---|---|---|---|---|---|---|---|
| Chr | Pos (hg38) | ID | Nearest Gene(s) | Annotation | Major/ Minor | AF Ca/Co | OR | *P*-Value | AF Ca/Co | OR | *P*-Value | OR | *P*-Value |
| 13 | 39111430 | rs1409440 | *NHLRC3;NXT1P1* | Downstream; Upstream | T/C | 0.17/ 0.09 | 2.03 | 2.12E-05 | 0.17/ 0.09 | 1.92 | 7.27E-02 | 2.01 | 3.98E-06 |
| 13 | 39134958 | rs7994616 | *NHLRC3;NXT1P1* | Downstream; Upstream | T/C | 0.17/ 0.09 | 2.03 | 2.15E-05 | 0.17/ 0.09 | 1.92 | 7.28E-02 | 2.01 | 4.00E-06 |
| 13 | 39140014 | rs11616618 | *NHLRC3;NXT1P1* | Downstream; Upstream | G/A | 0.17/ 0.09 | 2.03 | 2.32E-05 | 0.17/ 0.09 | 1.91 | 7.48E-02 | 2.00 | 4.48E-06 |
| 4 | 165159085 | rs9308098 | *TMEM192* | intronic | T/C | 0.21/ 0.12 | 1.87 | 2.79E-05 | 0.17/ 0.13 | 1.38 | 3.81E-01 | 1.78 | 2.50E-05 |
| 10 | 38680369 | rs11511253 | *SLC9B1P3* | intronic | G/A | 0.32/ 0.21 | 1.70 | 1.08E-04 | 0.30/ 0.22 | 1.53 | 1.61E-01 | 1.67 | 4.07E-05 |
| 2 | 145667963 | rs16825349 | *AC079163.1; AC079248.1* | Downstream; Upstream | A/G | 0.28/ 0.18 | 1.74 | 9.32E-05 | 0.24/ 0.19 | 1.37 | 3.20E-01 | 1.66 | 7.11E-05 |

List of the meta-analysis variants ($P_{meta}$ < 1 x $10^{-4}$ and same direction of effect for both studies, MAF ≥0.05) associated with lipoedema in discovery and replication studies. The variants have been annotated to their nearest genes. Allele Frequencies (AF), Odds Ratios (OR) and *P*-values are shown for both studies and meta-analysis. The variants have been sorted in ascending meta-analysis *P*-value order. Ca, cases; Co, controls.

associated either directly or through its LD buddies with the expression of the genes (*ZNF25*, *ZNF37A and ZNF33A*), pseudogenes (*HSD17B7P2*, *SEPT7P9*) and long non-coding RNA (*RP11-291L22.9/lnc-ZNF37A-4*) in, among others, lipoedema-related tissues like subcutaneous adipose tissue, and oestrogen-producing tissues such as adrenal gland and breast (S7 Table). rs9308098 ($P_{meta}$ = 2.50 x $10^{-5}$, $OR_{meta}$ = 1.79) is associated with *KLHL2* gene expression in adrenal gland tissue (S7 Table). While colocalization analysis does not provide support for all eQTL loci identified we observed evidence for colocalized GWAS and eQTL signals in adrenal gland, pancreatic and oesophageal muscular tissues implicating *LHFPL6* and *KLHL2* (SSP >3.2) (S8 Table; S2 Fig).

Further investigation of links between the top SNPs and the lipoedema phenotype was undertaken. Although likely underpowered by our modest sample size, a case only analysis of the discovery cohort identified no relationship between BMI and genotype of these top SNPs (P<0.01).

## Discussion

Lipoedema is a clinical diagnosis in urgent need of an understanding of mechanism and treatment. No good biomarkers exist, and the disease manifestations show phenotypic overlap with other disorders, hampering the clinical diagnosis. The cause of lipoedema remains elusive. It has been hypothesized that it is a form of obesity, a form of lipodystrophy/fat disorder, a hormonal disorder, a form of connective tissue disorder given the association with hypermobility and finally a lymphatic disorder given the frequent progression to lymphoedema [15, 39, 40]. Here we report the first comprehensive collection of lipoedema cases recruited from a white British population, with the aim of conducting a GWAS to explore a possible polygenic architecture. Through careful phenotyping we have been highly selective in recruitment of cases, taking care to exclude those with generalized obesity where lipoedema is difficult to diagnose.

We estimated SNP-based heritability of 50–60% in the discovery cohort, indicating a strong genetic link to lipoedema. However, larger lipoedema cohorts are needed to validate this estimation. Strong association of autosomal dominant inheritance with sex limitation has been

observed within affected family members with lipoedema [3]. Despite a small cohort size, we believe the careful phenotyping has led to the identification of some putative regions of genetic association. The top three SNPs in our analysis, rs1409440, rs7994616 and rs11616618, were located on chromosome 13 in a block of linkage disequilibrium (LD) close to the *LHFPL6* gene, with evidence for a colocalised eQTL implicating *LHFPL6* gene expression in lipoedema aetiology. The *LHFP (LHFPL6)* gene is a member of the lipoma HMGIC (High-mobility group protein isoform C) fusion partner gene family and it is localized to chromosome 13q. It has been associated with higher levels of polyunsaturated fats in adipose tissue in chicken thigh [41] and it has been linked to a translocation-associated lipoma [42], making it an interesting gene to explore. Petit *et al.* described an acquired cytogenetic translocation in a lipoma with breakpoints at 12q13-15 and 13q12 resulting in a fusion transcript between the genes *HMGIC* and *LHFP* [42]. Further cytogenetic analysis of various types of benign and malignant lipomas detected structural (balanced and unbalanced) rearrangements of or monosomy (clonal loss) for chromosome 13q and the authors speculated whether haplo-insufficiency was the pathogenetic mechanism [43].

Lipomas are common soft tissue tumours identified as a 'benign neoplasm of mature adipocytes' [44]. They are characterized by non-symmetrical fat accumulations which are soft, fatty lumps present in the subcutaneous layer. They have been reported in association with lipoedema [1, 45], and in our clinics some lipoedema patients have reported the presence of lipomas (see Fig 1D), but as they are not considered diagnostic of lipoedema, our data collection did not consistently record this. In contrast, lipomas are well described in Dercum's disease or "painful fat syndrome" [46] which lies within the spectrum of lipoedema. The localized deposits of fatty tissue around the knees seen in many individuals with lipoedema might represent lipoma-like adipose tissue [47, 48] and it has been suggested that lipoedema and lipomas may be associated as both can present with excessive adipose tissue [45]. The GWAS participants from our cohort carrying the risk alleles of the SNPs associated with *LHFPL6* were significantly more likely to report direct maternal family history compared to the non-carriers. However, how *LHFPL6* is linked to excessive adipose tissue in lipoedema is not known. Although further investigation is needed to prove causality in this correlation, this finding is consistent with a genetic association between this locus and the onset of familial lipoedema.

Lipoedema is often misdiagnosed as lymphoedema by inexperienced clinicians. However, it has been debated whether lymphatic dysfunction is a cause or result of lipoedema. This study did not identify any SNPs associated with genetic loci known to be involved in lymphatic development. This would corroborate the findings of Felmerer and colleagues [13] who reported that it is not a lymphatic phenotype underlying lipoedema. However, due to the limited size of our study, the possibility of such an association cannot be excluded.

Despite the strict selection criteria limiting numbers of recruits, the "UK Lipoedema" cohort is typical of other lipoedema cases described in the literature. The recruited lipoedema patients are strikingly similar to that of Dudek and colleagues, who reported similar low WHR ratios (average value = 0.78), self-reported high levels of easy bruising (91%), tenderness/pain (83%) and disproportional weight loss (87%) [17]. The age of onset was mainly reported as pubertal. The majority of women (86.7%) in our cohort reported disproportional weight loss upon dieting. However, it is important to also acknowledge these women reported that fat loss was achievable from affected limbs. Why so many women with lipoedema suffer with obesity is not yet understood–is it "cause and effect", or are there more complicated genetic reasons behind it? Clearly there is an urgent need for research into the possible association between lipoedema and obesity, but until then it is important to ensure that women with lipoedema access successful weight management strategies to ensure weight gain and progression of lipoedema are avoided.

Chronic fatigue, psychosocial and poor body image issues are recognized comorbidities with lipoedema. Many lipoedema patients will have been dismissed by their doctors at some point and told to manage their weight by dieting or lifestyle changes. Diets and physical exercise are reported to lead to disproportionate loss of weight from the upper half of the body in patients with lipoedema, accentuating the disproportional figure. The disproportionate body shape in females can cause negative body image and "body shaming" criticism from friends, family, and health care professionals. As an impact of overall psychological well-being this could lead to patient experiences of distress, anxiety, depression, eating disorders and isolation [2, 4]. The SF-36 questionnaire confirmed that quality of life was reduced in all eight domains evaluated. This is comparable to other studies of lipoedema patients using either the SF-36 or similar investigative tools [4, 22, 49]. The mean scores across eight domains show more consistency with chronic neuropathic pain patients than obesity patients (S3 Fig) [50, 51] suggesting similarities with individuals that have a chronic condition.

The main limitation of this study was the small numbers. We tried in particular to use BMI < 30 and WHR < 0.80 as inclusion criteria, but this resulted in too small a sample size. Thus, criteria had to be loosened to include cases with BMI ≤ 40 and WHR ≤ 0.85. This can only be recommended if there is sufficient medical history for the clinician to confirm the diagnosis. Lipoedema patients who present after onset of menopause were excluded. Lipoedema patients often present at times of hormonal change and that includes the menopause. If the latter group had been analysed, it is theoretically feasible that a different set of SNPs might have been uncovered, suggesting that the condition known as lipoedema might actually be a heterogeneous grouping of presentations with some biological mimicry. Despite being less conservative in inclusion, we still had a relatively small sample size, which limited our statistical power in the GWAS, but we believe the homogeneity of the cohort helped to enrich the dataset. Another limitation related to the samples obtained from the 'Understanding Society UK study' (controls) and GEL (cases and controls), which both lack information on waist-hip ratio and BMI. Such data would have been extremely valuable for excluding any potential lipoedema cases from the controls and to have understood if the cases from the GEL replication cohort would have fulfilled the lipoedema inclusion criteria.

In conclusion, we have described a tightly phenotyped lipoedema cohort from a UK population. Based on genetic analysis, we identified suggestive SNPs linked with the disease, notably at chr13q13.3 near the *LHFPL6* gene. The meta-analysis of the discovery and replication cohorts also revealed three other distinct genetic loci putatively associated with the disease. These results show some interesting connections relevant to the disease phenotype. However, replication of the GWAS in different populations is needed. From our findings, we cannot tell the true driver of disease and follow-up studies investigating the associated loci/genes are needed. In time this could enable a better understanding of the underlying genetic causes of lipoedema and its disease mechanism and perhaps even fat deposition and homeostasis in general.

## Supporting information

**S1 File. Supplementary methods.**
(PDF)

**S1 Table. Raw data collected for all 200 recruits in the 'UK Lipoedema' cohort.** Summarized in Table 2.
(XLSX)

**S2 Table. Raw SF-36 data for the 135 individuals who answered enough of the 36 questions to be included in the quality of life analysis.** Results are summarized in Table 3.
(XLSX)

**S3 Table. Correlations between SF-36 quality of life questionnaire domains.** The correlation coefficients, r, are displayed and those with strong correlations (r > 0.7) are highlighted in bold type. All correlations were significant at the p < 0.05 level.
(XLSX)

**S4 Table. GEL participants with lipoedema as "Recruited disease" in the rare diseases program of the 100,000 Genomes Project.** Age at recruitment is calculated as (year of recruitment to GEL)–(Year of Birth). Family history is based on any reports of "Affected" family members in GEL; ".", uncertain.
(XLSX)

**S5 Table. List of the top 26 variants in the discovery study, and their GWAS results in the replication study and meta-analysis.** The variants have been annotated to their nearest genes. Effect sizes and P values are shown for both studies and meta-analysis. Direction column shows whether the discovery and replication study follow the same direction of effect.
(XLSX)

**S6 Table. List of Genome regulatory elements associated with the three top meta-analysis SNPs (rs1409440, rs7994616, rs11616618) on chromosome 13 potentially associated with lipoedema.** Data was downloaded from UCSC Table Browser using the geneHancerClusteredInteractionsDoubleElite (last updated: 2019-01-15) and encodeCcreCombined (last updated: 2020-05-20) tables.
(XLSX)

**S7 Table. List of all gene expression quantitative trait loci found for the SNPs in Table 4 and/or their LD buddies (P < 0.001, $r^2$ > 0.6, European Population).** Analysis was performed on LDexpress module from the LDlink online tool of NCBI and this list was downloaded.
(XLSX)

**S8 Table. List of all colocalized signals from LocusFocus (SSP >0) for the 6 replicated loci in Table 4.** Strong evidence signals for colocalization (SSP > 3.2, Bonferroni correction for 17 gene-tissue pairs tested for colocalization) are highlighted in bold face.
(XLSX)

**S1 Fig. Distribution of BMI, WHR and waist circumference.** Of the 200 lipoedema cases recruited to the 'UK Lipoedema' cohort, we have anthropometric data for 161 (A and C). 130 individuals of white British descent were selected for GWAS of which 105 have been plotted in (B and D). (A, B) Waist circumference vs BMI show that many individuals fall in the overweight (BMI over 25 kg/m2; yellow line) and obese (BMI over 30 kg/m2; red line) categories. According to the NHS waist measurement guidelines for white European women, a waistline < 80cm is low risk (in green), high risk 80–88cm (in blue) and very high risk > 88cm (in red) of developing diseases such as type 2 diabetes, hypertension, coronary heart disease, cancer and stroke [1]. (C, D) Waist-hip ratio (WHR) vs BMI show that the majority of cases included in the study have a WHR < 0.85, which according to WHO guidelines is healthy [2]. Any cases outside the region of inclusion, i.e. the cases with a WHR > 0.85 (and BMI > 40), have been carefully investigated by the clinicians involved before being included in the study (see Supplementary Methods, S1 File, for details on case ascertainment).
(PDF)

**S2 Fig. Plot of the SNPs showing evidence of colocalization from LocusFocus.** Heatmap shown summarize the SS colocalization tests for all the genes in the ±500 kb region of each SNP across all 54 GTEx tissues. Strength of colocalization is coloured from yellow (low *-log10 (P)*) to red (high *-log10(P)*). White indicates either no eQTL data or gene-tissue pair does not have significant eQTL signal after Bonferroni correction. **A**. Heatmap shows results where SSP> 0 for genes in the region around **rs1409440** illustrating the colocalization of *LHFPL6* eQTLs across multiple tissues. **B**. Heatmap shows results where SSP>0 for genes in the region around **rs9308098** illustrating colocalization of the *KLHL2* eQTL in adrenal tissue. (PDF)

**S3 Fig. Radar diagram showing the mean score for each of the 8 domains from the SF-36 quality of life questionnaire in lipoedema patients taken from Table 3 (red).** The lipoedema cases are comparatively similar to patients with chronic neuropathic pain (blue, SF36 data taken from Torrance *et al.* [3]). In contrast, overweight female without lipoedema (yellow, Sahle *et al.* [4]) are similar on many domains to healthy weight female (light green, Sahle *et al.* [4]; or dark green, Bowling *et al.* [5]), whilst obese female without lipoedema (orange, Sahle *et al.* [4]) show a lower score on some domains but not as low as the lipoedema cases. The healthy females from Bowling *et al.* (dark green; [5]) are age matched to the lipoedema cases, whereas the data from Sahle *et al.* [4] include males. The data on patients with chronic pain were taken from a general population of over 18 years old attending their GP service [3]. (PDF)

## Acknowledgments

The authors thank all participants for volunteering their time for this study. We would also like to thank 'LipoedemaUK' for facilitating recruitment through their members. This research was made possible through access to the data and findings generated by the 100,000 Genomes Project and Understanding Society. The 100,000 Genomes Project is managed by Genomics England Limited (a wholly owned company of the Department of Health and Social Care) and uses data provided by patients and collected by the National Health Service as part of their care and support (Genomics England Consortium). Understanding Society is under the scientific leadership of the Institute for Social and Economic Research, University of Essex, and survey delivery by NatCen Social Research and Kantar Public. The research data are distributed by the UK Data Service. We also extend our thanks to members of the St George's University of London (SGUL) Lymphovascular Research Team for invaluable discussions and feedback on our work and to the following members of the Lipoedema consortium: Dr Yann Klimentidis, University of Arizona; Prof Natasha Harvey and Dr Hamish Scott, University of South Australia.

## Author Contributions

**Conceptualization:** Steve Jeffery, David B. Savage, Peter S. Mortimer, Vaughan Keeley, Kristiana Gordon, Pia Ostergaard.

**Data curation:** Dionysios Grigoriadis, Ege Sackey, Sara E. Dobbins.

**Formal analysis:** Dionysios Grigoriadis, Malou van Zanten, Mike Mills, Sara E. Dobbins, Li Ling Lee, Alan Pittman, Pia Ostergaard.

**Funding acquisition:** Steve Jeffery, Peter S. Mortimer, Kristiana Gordon, Pia Ostergaard.

**Investigation:** Dionysios Grigoriadis, Ege Sackey, Katie Riches, Malou van Zanten, Liang Dong, Alan Pittman, Kristiana Gordon.

**Methodology:** Dionysios Grigoriadis, Alan Pittman.

**Project administration:** Glen Brice, Vaughan Keeley, Kristiana Gordon, Pia Ostergaard.

**Resources:** Katie Riches, Malou van Zanten, Glen Brice, Ruth England, David B. Savage, Peter S. Mortimer, Vaughan Keeley, Kristiana Gordon, Pia Ostergaard.

**Software:** Dionysios Grigoriadis, Alan Pittman.

**Supervision:** Vaughan Keeley, Alan Pittman, Kristiana Gordon, Pia Ostergaard.

**Validation:** Sara E. Dobbins, Kristiana Gordon, Pia Ostergaard.

**Visualization:** Dionysios Grigoriadis, Ege Sackey, Katie Riches, Malou van Zanten, Mike Mills, Li Ling Lee, Pia Ostergaard.

**Writing – original draft:** Dionysios Grigoriadis, Ege Sackey, Malou van Zanten, Pia Ostergaard.

**Writing – review & editing:** Dionysios Grigoriadis, Ege Sackey, Katie Riches, Malou van Zanten, Glen Brice, Ruth England, Mike Mills, Sara E. Dobbins, Li Ling Lee, Steve Jeffery, Liang Dong, David B. Savage, Peter S. Mortimer, Vaughan Keeley, Alan Pittman, Kristiana Gordon, Pia Ostergaard.

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
