## [Decision Letter · Decision Letter 0]

14 Sep 2021

PONE-D-21-18905Investigation of clinical characteristics and genome associations in the 'UK Lipoedema' cohortPLOS ONE

Dear Dr. Ostergaard,

Thank you for submitting your manuscript to PLOS ONE. After careful consideration, we feel that it has merit but does not fully meet PLOS ONE’s publication criteria as it currently stands. Therefore, we invite you to submit a revised version of the manuscript that addresses the points raised during the review process. Please submit your revised manuscript by Oct 29 2021 11:59PM. If you will need more time than this to complete your revisions, please reply to this message or contact the journal office at plosone@plos.org. Please include the following items when submitting your revised manuscript:A rebuttal letter that responds to each point raised by the academic editor and reviewer(s). You should upload this letter as a separate file labeled 'Response to Reviewers'.A marked-up copy of your manuscript that highlights changes made to the original version. You should upload this as a separate file labeled 'Revised Manuscript with Track Changes'.An unmarked version of your revised paper without tracked changes. You should upload this as a separate file labeled 'Manuscript'.

We look forward to receiving your revised manuscript.

Kind regards,

Junwen Wang, Ph.D.

Academic Editor

PLOS ONE

“The 100,000 Genomes Project is funded by the National Institute for Health Research and NHS England. The Wellcome Trust, Cancer Research UK and the Medical Research Council have also funded research infrastructure.”

“This project was supported by Lipedema Foundation (https://www.lipedema.org/) LF#006 (KG and PO), the Wellcome Trust (https://wellcome.org/) WT107064 (DBS), and the MRC Metabolic Disease Unit, and The National Institute for Health Research (NIHR) Cambridge Biomedical Research Centre and NIHR Rare Disease Translational Research Collaboration (https://www.mrl.ims.cam.ac.uk/mrc-metabolic-diseases-unit/) MRC_MC_UU_12012.1 (DBS). The funders had no role in study design, data collection and analysis, decision to publish, or preparation of the manuscript.”

4. We note that you have stated that you will provide repository information for your data at acceptance. Should your manuscript be accepted for publication, we will hold it until you provide the relevant accession numbers or DOIs necessary to access your data. If you wish to make changes to your Data Availability statement, please describe these changes in your cover letter and we will update your Data Availability statement to reflect the information you provide

5. Please amend the manuscript submission data (via Edit Submission) to include author Lipoedema Consortium, Genomics England Research Consortium^4^

Additional Editor Comments (if provided):

Reviewers' comments:

Reviewer's Responses to Questions

**Comments to the Author**

1. Is the manuscript technically sound, and do the data support the conclusions?

Reviewer #1: Yes

Reviewer #2: Yes

2. Has the statistical analysis been performed appropriately and rigorously? 

Reviewer #1: Yes

Reviewer #2: I Don't Know

3. Have the authors made all data underlying the findings in their manuscript fully available?

Reviewer #1: No

Reviewer #2: Yes

4. Is the manuscript presented in an intelligible fashion and written in standard English?

Reviewer #1: Yes

Reviewer #2: Yes

5. Review Comments to the Author

Reviewer #1: The article provides insights into the genetics of Lipedema that will potentially aid in defining the pathogenesis of the disease.

Comments:

1. Have the authors any information on the stage of Lipedema patients? It would be very interesting to correlate the SNP data with the different stages if possible. The authors might find the article by Herbst et al. helpful for the lipedema diagnosis (https://pubmed.ncbi.nlm.nih.gov/34049453/)

2. Have the authors considered analyzing the data between upper and lower limb data separately?

3. The authors mentioned that SNPs might be associated with the biosynthesis of the hormones, Do the authors have any information on the hormonal status of the patients and/or the controls?

4. The inclusion criteria included patients with BMI <40 with no abdominal Obesity. However, the range of BMI (19-58.8) is very wide. Have the authors considered analyzing the data based on the BMI classes (overweight vs obese)?

5. The authors mentioned that LHFPL6 as a potential marker for Lipedema. Since not all lipedema patients have lipomas, have the authors considered analyzed the data by grouping non-lipoma vs lipoma lipedema patients?

Reviewer #2: In their Study, Grigoriadis et al perform the first genome association study on lipedema patients, aiming to unravel the genetic background of the disease and allocate potential genes to lipedema that is still lacking biomarkers whereas the molecular background remains elusive.

While of great interest, the study requires extensive clarifications and further analysis of the already acquired data to strengthen the conclusions and outcomes of the analysis. A detailed list of the questions to be answered is below:

- What kind of physician set the diagnosis? Has an angiologist been involved to rule out presence of lymphedema or impaired lymphatic function? If yes, have any specific tests been undertaken for this purpose?

- Please present the stage of lipoedema for the patients included in the analysis - this is not mentioned in the suppl. material

- In Table 2 it is indicated that 39.1% of patients included are severely obese – has a clustering based on BMI taken place? This would be advisable, even if the BMI of the control databases is not clear. What is more, patients with oedema of the feet should be either excluded or analysed separately (probably lymphedema is present). This

- In regard to bruising: commonly the patient reported bruising deviates from the clinically observed at the time of the clinical examination. What was the % of bruising present during the clinical assessment?

- Has analysis of specific clusters taken place? In particular analysis of SNPs of specific lipoedema stage. This is essential to understand the dynamics of the disease, particularly given Stage IV (lipolymphoedema) is very much clinically different that Stage I-III lipoedema

- Previous articles published by other groups have already suggested potential gene involvement. These articles are neither cited nor discussed. Please include in your discussion. Articles: Felmerer et al J Surg Res (DOI: 10.1016/j.jss.2020.03.055) and Felmerer et al Sci Rep (DOI: 10.1038/s41598-020-67987-3)

- Whether lymphatic vascular involvement is a cause or results of lipoedema is still quite debatable. The authors are requested to discuss this aspect based on the SNPs they have identified

6. PLOS authors have the option to publish the peer review history of their article (what does this mean?). If published, this will include your full peer review and any attached files.

Reviewer #1: No

Reviewer #2: No

---

## [Author Response · Author response to Decision Letter 0]

26 Oct 2021

See attached 'Response to Review' for our comments to all three Reviewers.

---

## [Decision Letter · Decision Letter 1]

16 Mar 2022

PONE-D-21-18905R1Investigation of clinical characteristics and genome associations in the 'UK Lipoedema' cohortPLOS ONE

Dear Dr. Ostergaard,

Thank you for submitting your manuscript to PLOS ONE. After careful consideration, we feel that it has merit but does not fully meet PLOS ONE’s publication criteria as it currently stands. Therefore, we invite you to submit a revised version of the manuscript that addresses the points raised during the review process.

While the study is of interest, there are major issues related to the GWAS analysis and its interpretation (see Reviewer 5's comments) that need to be addressed for the manuscript to be acceptable for publication.

We look forward to receiving your revised manuscript.

Kind regards,

Dylan Glubb

Academic Editor

PLOS ONE

Reviewers' comments:

Reviewer's Responses to Questions

**Comments to the Author**

1. If the authors have adequately addressed your comments raised in a previous round of review and you feel that this manuscript is now acceptable for publication, you may indicate that here to bypass the “Comments to the Author” section, enter your conflict of interest statement in the “Confidential to Editor” section, and submit your "Accept" recommendation.

Reviewer #4: (No Response)

Reviewer #5: (No Response)

Reviewer #6: (No Response)

2. Is the manuscript technically sound, and do the data support the conclusions?

Reviewer #4: Yes

Reviewer #5: No

Reviewer #6: Yes

3. Has the statistical analysis been performed appropriately and rigorously? 

Reviewer #4: Yes

Reviewer #5: No

Reviewer #6: Yes

4. Have the authors made all data underlying the findings in their manuscript fully available?

Reviewer #4: Yes

Reviewer #5: No

Reviewer #6: Yes

5. Is the manuscript presented in an intelligible fashion and written in standard English?

Reviewer #4: Yes

Reviewer #5: Yes

Reviewer #6: Yes

6. Review Comments to the Author

Reviewer #4: Its very interesting study which involve more than 130 patients genetic data. Study developed some new knowledge about disease which will be helpful for future disease description.

Reviewer #5: Is there an estimate of the prevalence of lipoedemia in the population? This would be useful to be added to the introduction. If there is no estimate then that should be discussed? Do the authors have an idea of what the incidence is based on their recruitment of cases? Or if the incidence is rising in line with rising rates of obesity?

This is just a comment, but lipedema appears to be quite rare and I wonder if a GWAS approach is appropriate for this disease. It is hypothesized that autosomal dominant variants are likely to cause lipoedemia (i.e. rare frequency variants, PMID: 20358611), so would suggest sequencing studies of well-phenotyped, familial samples would be a suitable approach to identify genetic risk factors. The authors mentioned that no monogenic cause has been identified for lipedema but I’m unsure if this is because these sequencing studies have not been performed?

Most of my comments relate to the GWAS methodology and result reporting.

Were duplicate samples assessed for genotyping (not duplicates between the two batches, but duplicates assessed in the same batch)? This should be performed as standard for genotyping studies and SNPs which are not concordant (e.g. concordance <98%) between duplicates removed.

As the study includes only female participants, was X-chromosome heterozygosity assessed to investigate whether there were any suspected male genotypes, XO or XXY individuals?

Was there a reason imputation was not performed for the study?

The authors have used a MAF>0.01 as a quality control cut-off. However, given the very small case numbers included in this study, I strongly recommend a MAF>0.05 be used. Very large odds ratios, particular those >3, are not compatible with a common SNP-disease model. The variants which have large ORs in this study (2 SNPs OR> 3, 1 SNP OR>2.5) all have MAF<0.05 in controls, and I would consider the estimates for these variants to be unreliable.

What does “suggestive” mean in terms of suggestive genomic loci? What is the cut-off for calling something suggestive?

When assessing whether a SNP affects expression of a gene, the eQTL signal should be assessed for colocalization with the phenotype signal. The latest GTEx analysis found that >90% of SNPs were associated with expression of at least one gene, in at least one tissue using a nominal P-value <0.05. Doing a simple look-up and using an arbitrary P-value cut-off for eQTL assessment is not good practice.

Have the identified loci in this study been identified as associated with other related traits by GWAS (e.g. waist-hip ratio, body proportions – e.g. leg/arm/trunk fat ratio, steroid hormone levels etc)?

The study cohort is way too small to perform case-only analysis of the relationship between BMI and genotype. I recommend this is removed and mentioned only as a limitation of the study that this type of analysis could not be performed.

“Approximately half the recruited women reported a family history of large legs, and this is consistent with the estimated SNP-based heritability of 50-60% calculated in the discovery cohort, indicating a strong genetic link to lipoedma.” This sentence is incorrect and should be deleted. Are the authors suggesting that the entire familial heritability of lipoedema to due to common genetic variation? I would suggest this is not the case, as with most complex diseases, heritability would be due to a number of factors including low-frequency, high-risk variants and environmental risk factors. Importantly, the SNP heritability is unreliable with very large standard errors – SEs are, in fact, larger than the SNP heritability estimates. Related to this, the authors have used prevalence estimates of 5% and 10% in their SNP heritability estimates but have not provided justification for this.

The authors mention that “fine-mapping analysis results showed…” however, there was no indication in the manuscript that fine-mapping was performed? What do the authors mean by fine-mapping?

The authors also state “finding from our enrichment analysis”, but then proceed to discuss eQTL look-up results. This is not an enrichment analysis.

Given the relationship between obesity and lipedema, it could be useful to construct polygenic scores for obesity measures in your cohort and assess whether these are significantly different between cases and controls.

Can lipedema cases be identified from the UK Biobank cohort? These could be used as another independent replication set, even if not as well-phenotyped as your discovery set.

I highly recommend that summary statistics for GWAS is submitted to the GWAS Catalog to be accessed by the scientific community.

Reviewer #6: The paper is one of the first to describe a GWAS analysis on a very important health issue affecting predominantly women, lipodema. It well written and the bioinformatics relating to the study design is robust. To that end, I have some questions:

1. The authors describe a ‘strict’ selection criteria on how the patients were recruited. As they are aware, the cohort numbers and choice is critical to a meaningful outcome. What precautions were taken to exclude patients with any underlying health conditions such as thyroid problems, vascular defects, cardiovascular, cancer, diabetes. There is a very large variation with BMI- how did the authors select the non lipodema cohort? What criteria was applied here- were they BMI matched age matched controls? If so, then are they comparing lipodema to obesity? With regards to the upper body, what criteria was used to ascertain minimal fat- callipers and fold test? What would the cut off be for this if minimal upper body obesity was used as a selection criteria?

2. How were the patients diagnosed with lymphoedema? Were the patients lymphatics measured by lymphosintigraphy? Did the authors identify any SNPs in the patient with venous issues such as Sox18? Do the authors believe that lymphoedema is a secondary side effect of excessive fat formation rather than the cause or an initiator of lipodema? There was a significant degree of oedema and bruising in patients- did the authors identify any SNPs in VegfA related pathways?

3. The SNP identified as key in lipoma is interesting. How many of the cohort patients were diagnosed with lipoma or dercums disease? Was this used in the selection criteria as a possible way to segment the cohort?

4. Have any of the SNPs been validated? I would like to see a few of these validated by Sanger to strengthen the paper. This would be a simple experiment.

5. Can the authors speculate about the size of the cohort that would be needed to provide a statistical confidence in the analysis.

7. PLOS authors have the option to publish the peer review history of their article (what does this mean?). If published, this will include your full peer review and any attached files.

Reviewer #4: **Yes: **Musarat Ishaq

Reviewer #5: No

Reviewer #6: No

---

## [Author Response · Author response to Decision Letter 1]

31 May 2022

We have responded to the reviewer comments in the uploaded file "PONE-D-21-18905R2_Response to Review"

---

## [Decision Letter · Decision Letter 2]

5 Jul 2022

PONE-D-21-18905R2Investigation of clinical characteristics and genome associations in the 'UK Lipoedema' cohortPLOS ONE

Dear Dr. Ostergaard,

Thank you for submitting your manuscript to PLOS ONE. After careful consideration, we feel that it has merit but does not fully meet PLOS ONE’s publication criteria as it currently stands. Therefore, we invite you to submit a revised version of the manuscript that addresses the points raised during the review process. The authors particularly need to address the comments from Reviewer #5 before the manuscript can be accepted. As the reviewer points out, the associations for the low frequency alleles are highly likely to be spurious with such low case numbers and a MAF threshold of at least 5% should be used. With regards to Reviewer #6's suggestion to validate SNPs by Sanger sequencing. this is not typically performed for GWAS and is not necessary for acceptance of the manuscript.

We look forward to receiving your revised manuscript.

Kind regards,

Dylan Glubb

Academic Editor

PLOS ONE

Journal Requirements:

Reviewers' comments:

Reviewer's Responses to Questions

**Comments to the Author**

1. If the authors have adequately addressed your comments raised in a previous round of review and you feel that this manuscript is now acceptable for publication, you may indicate that here to bypass the “Comments to the Author” section, enter your conflict of interest statement in the “Confidential to Editor” section, and submit your "Accept" recommendation.

Reviewer #5: (No Response)

Reviewer #6: (No Response)

2. Is the manuscript technically sound, and do the data support the conclusions?

Reviewer #5: Partly

Reviewer #6: Partly

3. Has the statistical analysis been performed appropriately and rigorously? 

Reviewer #5: No

Reviewer #6: Yes

4. Have the authors made all data underlying the findings in their manuscript fully available?

Reviewer #5: Yes

Reviewer #6: Yes

5. Is the manuscript presented in an intelligible fashion and written in standard English?

Reviewer #5: Yes

Reviewer #6: Yes

6. Review Comments to the Author

Reviewer #5: The authors have done a good job in addressing my previous comments. I just have a few further queries.

I assume the authors have a typo answering my question regarding imputation – that the cases and controls were NOT genotyped on the same platform. This is a hurdle, but could be easily rectified by harmonizing the two groups to the same SNP lists before imputation. I don’t see that the smaller case size would have an impact on the reliability of the imputation. I don’t expect the authors to impute for this study, but would recommend for future work.

I still strongly recommend that SNPs that a MAF > 5% is used for the study. For a sample of 148 cases, only 7-8 cases need to be carriers for testing. The low allele frequencies, high ORs and wide confidence intervals suggest these results to be likely driven by statistical artefact. As previous, the ORs for these variants are too high to be compatible with a common variant model. I also note that the table doesn’t provide CIs and a table footnote is not enough to warn readers for interpretation.

I appreciate that the authors have performed colocalization analyses. There should be futher information provided, including which GTEx dataset was used (v8 or v7?). Given the large number of genes and tissues assessed, the multiple testing burden would have been high. Could the authors provide what their cut-offs were in light of this (for eQTL p-value and colocalization SSP)? It looked like there was evidence of colocalization for rs10499948 and LHFP in pancreas (ST8) but this wasn’t in fig S7, why was this? Also there was some evidence for LHFP in liver in fig S7 but this wasn’t in ST8. Why?

Reviewer #6: The authors should validate the a selection of SNPs identified to ensure the conclusion is sound. It is not clear that the normal cohort is classified as obese or not. This could potentially impact the findings.

7. PLOS authors have the option to publish the peer review history of their article (what does this mean?). If published, this will include your full peer review and any attached files.

Reviewer #5: No

Reviewer #6: No

---

## [Author Response · Author response to Decision Letter 2]

16 Aug 2022

Response to 3rd review has been attached.

---

## [Editor Report · Decision Letter 3]

7 Sep 2022

Investigation of clinical characteristics and genome associations in the 'UK Lipoedema' cohort

PONE-D-21-18905R3

Dear Dr. Ostergaard,

We’re pleased to inform you that your manuscript has been judged scientifically suitable for publication and will be formally accepted for publication once it meets all outstanding technical requirements.

Kind regards,

A/Prof Dylan Glubb

Academic Editor

PLOS ONE

Additional Editor Comments (optional):

While the authors have finally addressed all the reviewers' comments and the paper can be provisionally accepted, there is some lack of clarity that needs to be addressed before publication:

1. The abstract should provide some information about the level of association of rs1409440 (e.g. OR, CIs, p-value, etc.) rather than just calling it the top SNP.

2. In lines 350-353, there is discussion of patients who carry the LHFPL6 'SNPs'. Presumably the authors are referring to patients carrying the risk alleles of these SNPs - this should be made clear.

3. It should be stated in the main text in which tissues the GWAS variants/eQTLs colocalised, given that these findings are much more likely to be causal than the other eQTLs discussed.
---

## [Editor Report · Acceptance letter]

3 Oct 2022

PONE-D-21-18905R3 

Investigation of clinical characteristics and genome associations in the ‘UK Lipoedema’ cohort 

Dear Dr. Ostergaard:

I'm pleased to inform you that your manuscript has been deemed suitable for publication in PLOS ONE. Congratulations! Your manuscript is now with our production department. 

Kind regards, 

on behalf of

Dr. Dylan Glubb 

Academic Editor

PLOS ONE